# Method to measure the size-resolved real part of aerosol refractive index using differential mobility analyzer in tandem with single particle soot photometer

Gang Zhao[1], Weilun Zhao[1], Chunsheng Zhao[1*]

1 Department of Atmospheric and Oceanic Sciences, School of Physics, Peking University, Beijing, China

*Correspondence to: Chunsheng Zhao (zcs@pku.edu.cn)*

**Abstract**

Knowledge on the refractive index of ambient aerosol can help reduce the uncertainties in estimating aerosol radiative forcing. A new method is proposed to retrieve the size-resolved real part of refractive index (RRI). Main principle of deriving the RRI is measuring the scattering intensity by single particle soot photometer of size-selected aerosol. This method is validated by a series of calibration experiments using the components of known RRI. The retrieved size-resolved RRI cover a wide range from 200 nm to 450 nm with uncertainty less than 0.02. Measurements of the size resolved RRI can improve the understanding of the aerosol radiative effects.

**1 Introduction**

Aerosols exert significant influence on the earth energy budget by scattering and absorbing radiation (Ramanathan and Carmichael, 2008). There still remain great uncertainties when estimating the aerosol effective radiative forcing (RF) (Ghan and Schwartz, 2007) and an accurate estimation of the aerosol optical properties can help reduce the RF variations. The optical properties of the ambient aerosol particles are determined by their particle size and complex refractive index (RI, $m=n+ki$) (Bohren and Huffman, 2007;Levoni et al., 1997). Despite that the ambient aerosol particle size distribution can be measured with high accuracy (Wiedensohler et al., 2012), an accurate measurement of the ambient aerosol RI remains challenging. The RI is also widely used in remote sensing (Redemann et al., 2000;Dubovik, 2002;Zhao et al., 2017) and atmospheric modelling (Ghan and Schwartz, 2007;Kuang et al., 2015) because the aerosol single scattering albedo (SSA) and aerosol scattering phase function are highly related with the RI. At the same time, a small uncertainty in the real part of the RI (RRI) can lead to great uncertainties when estimating the aerosol RF. Zarzana et al. (2014) found that a variation of 0.003 in RRI can lead to uncertainties of 1% in RF for non-absorbing ammonium sulfate particles. Moise et al. (2015) estimated that the RF would increase 12% when the

RRI varied from 1.4 to 1.5. Valenzuela et al. (2018) reported that the uncertainties in RF is estimated
to be 7% when the aerosol RRI varied by 0.1. Therefore, it is pressing that the uncertainties of the RI
be reduced when estimating the RF.
Many methods were proposed to derive the RRI. The RRI can be estimated by linear volume
average of the known aerosol chemical components by
$$\mathrm{n} = \sum_i f_i \, n_i \qquad (1)$$

where $f_i$ and $n_i$ is the volume fraction and known partial refractive index of $i$th component (Wex et
al., 2002;Hand and Kreidenweis, 2002;Hänel, 1968;Liu and Daum, 2008). The aerosol RRI can also
be calculated by partial molar refraction approach (Stelson, 1990;Hu et al., 2012) which is essentially
consistent with the linear volume method (Liu and Daum, 2008). The ambient aerosol RRI can be
derived by synthetically using the radiative transfer calculations and the ground-based solar extinction
and scattering measurements (Wendisch and Hoyningen-Huene, 1994, 1992). Sorooshian et al. (2008)
developed a method to measure the aerosol RRI based on the differential mobility analyzer (DMA)
and an optical particle counters. The RRI could be retrieved from the known particle size from the
DMA and the aerosol scattering intensity from the Optical Particle Counter (OPC) for aerosol particles
larger than 500 nm. The Scanning Mobility Particle Sizer (SMPS) and OPC was used in combination
to derive the RRI by aligning the particle size distributions in the instrument overlap regions (Hand
and Kreidenweis, 2002;Vratolis et al., 2018). The aerosol effective RRI was also retrieved by applying
Mie scattering theory to the aerosol particle number size distribution, aerosol bulk scattering
coefficient and aerosol absorbing coefficient data (Cai et al., 2011;Liu and Daum, 2000). Spindler et
al. (2007) retrieved the aerosol RRI value by using the cavity ring-down spectroscopy to measuring
the scattering and absorbing properties of bulk aerosols. Eidhammer et al. (2008)) measured the light
scattering at different angles and retrieved the RRI. Similarly, the aerosol RRI was retrieved by
measuring the aerosol phase function (Barkey et al., 2007). Recently, a method by using the single
particle mass spectrometry was proposed to measure the aerosol RRI (Zhang et al., 2015). At the same
time, aerosol time-of-flight mass spectrometer was proved to be capable of measuring the aerosol RRI
(Moffet et al., 2008). The aerosol RRI can also be retrieved from the Mie spectroscopy by using the
optical tweezers in the laboratory (Shepherd et al., 2018).
Many studies show that aerosol of different diameters share different properties such as shape
(Zhang et al., 2016;Peng et al., 2016), density (Qiao et al., 2018), aerosol hygroscopicity (Wang et al.,

2017) and most importantly, the chemical components (Liu et al., 2014;Hu et al., 2012). Thus, there might be significant variations in the aerosol RRI for aerosols of different diameters because the aerosol RRI is highly related to the aerosol density (Liu and Daum, 2008) and chemical components (Stelson, 1990). On the other way round, information of the size-resolved aerosol RRI can help to study the chemical information and the aging process of aerosols among different diameters. Therefore, measurement of the size-resolved aerosol RRI is necessary.

Up to now, there are few information in the literature of the size-resolved ambient aerosol RRI (Ebert et al., 2004;Kandler et al., 2007;Ebert et al., 2002). Traditionally, the size-resolved ambient aerosols RRI are estimated by measuring the molar fraction or volume fraction of main aerosol chemical compositions. However, the influence of organic component on the aerosol RRI is ignored. The organic component contributes more than 20% of the total aerosol component in the North China Plain (Hu et al., 2012;Liu et al., 2014). At the same time, RRI of the organic aerosol changes significantly between 1.36 and 1.66 (Moise et al., 2015). Ignoring the organic component may lead to significantly uncertainties when estimating the aerosol RRI. There were no technique, to our knowledge, that directly measures the size-resolved aerosol optical properties and derives the size-resolved aerosol RRI.

In this study, a novel method is proposed to measure the size-resolved ambient aerosol RRI by using a DMA in tandem with a single particle soot photometer (SP2). The principle of the system is using the SP2 to measure the scattering properties of size-selected aerosols. Knowing the aerosol diameter and corresponding scattering intensity, the size-resolved aerosol RRI can be retrieved based on the Mie scattering theory. This proposed method can measure the ambient aerosol RRI over a wide size range with high accuracy. The measurement system is employed in a field campaign in the North China Plain and the corresponding results are further discussed.

The structure of this manuscript is as follows: section 2 provides the instruments setup and details of the instrument. The method to retrieve the size-resolved aerosol RRI is given in section 3. Section 4 shows the discussions about the uncertainties of the proposed method and field measurements results of the size-resolved aerosol RRI. Conclusions come at the last part.

**2 Instrument**

**2.1 Instrument Setup**

The instrument setup is schematically shown in fig. 1(a). Firstly, the dried sample aerosols are
guided to a X-ray soft diffusion charger and then lead to a DMA (Model 3081, TSI, USA). The quasi-
monodisperse aerosols that pass though the DMA at a given diameter are then drawn into a SP2 to
measure the aerosol scattering properties with a flow ratio of 0.12 lpm and a condensation particle
counter (CPC, Model 3776, TSI, USA) to count the aerosol number concentration with a flow ratio of
0.28 lpm respectively. Thus, the sample flow ($Q_a$) of the DMA is 0.4 lpm. Accordingly, the sheath
flow ($Q_{sh}$) of the DMA is 4 lpm. The DMA is set to scan the aerosols diameter from 12.3 to 697 nm
over a period of 285s and repeats after a pause of 15 s. The combination of DMA, CPC and SP2 can
provide information of aerosol particle number size distribution (PNSD) and size resolved RRI.
On 8$^{th}$, June, 2018, the measurement system was employed at the filed measurement of
AERONET station of BEIJING_PKU (N39°59′, E116°18′) to test the reliability of retrieving the
ambient size-resolved RRI. This measurement site locates on the north west of the city of Beijing,
China and is about 1.8 km north of the Zhongguancun, Haidian District, which is one of the busiest
areas in Beijing. It is surrounded by two main streets: Zhongguancun North Street to the west and
Chengfu Road to the south. This site can provide representative information of the urban roadside
aerosols (Zhao et al., 2018).
**2.2 DMA**
When a voltage (*V*) is applied to the DMA, only a narrow size range of aerosol particles, with the
same electrical mobility ($Z_p$) can pass through the DMA (Knutson and Whitby, 1975). The $Z_p$ is
expressed as:
$$Z_P = \frac{Q_{sh}}{2\pi VL} ln(\frac{r_1}{r_2})$$    (2)
where $Q_{sh}$ is the sheath flow rate; *L* is the length of the DMA; $r_1$ is the outer radius of annular space
and $r_2$ is the inner radius of the annular space. The transfer function refers to the probability that a
particle with a certain electrical mobility can pass through the DMA. For a given *V*, the transfer
function is triangular-shaped, with the peaking value of 100% and a half width (HW) of
$$\Delta Z_p = Z_P \frac{Q_a}{Q_{sh}}$$    (3)
The aerosol $Z_p$, which is highly related to the aerosols diameter ($D_p$) and the number of elementary
charges on the particle (*n*), is defined as:
$$Z_p = \frac{neC(D_p)}{3\pi\mu D_p} \qquad (4)$$

where $e$ is the elementary charge; $\mu$ is the gas viscosity coefficient, $C(D_p)$ is the Cunningham slip
correction that is defined by:
$$C = 1 + \frac{2\tau}{D_p}(1.142 + 0.558e^{-\frac{0.999D_p}{2\tau}}) \qquad (5)$$

where $\tau$ is the gas mean free path.
Based on the discussion above, the aerosols that pass through the DMA with the same $Z_p$, can
have different $D_p$ and different elementary charges.
**2.3 SP2**
The SP2 is a widely used instrument that can measure the optical properties of every single particle.
The measurement principle and instrumental setup of the SP2 have been discussed in detail previously
(Stephens et al., 2003;Schwarz et al., 2006) and will be briefly described here. When the sample
aerosol particles pass through the continuous Nd:YAG laser beam at 1064 nm with the circulating
power about 1 mW/cm$^2$ in the cavity, eight sensors distributed at four directions are synchronously
detecting the emitted or scattered light by using avalanche photo-detector (APD) at different angles
(45° and 135°). For each direction, the two APDs sample the same signal with different sensitivities to
get a wider measurement range. The low gain channels are less sensitive to the measured signal and
can be used to measure the stronger signal of larger particles. In accordance, the high gain channels
are more sensitive to the measured signal, and can be used to measure the weaker signal of smaller
particles. The optical head of the SP2 is shown schematically in fig. 1(b).
In this study, we utilize signals from four channels of the SP2: two of them measure the scattering
signals and another two measure the incandescent light between 350 nm and 800 nm. The peak height
(*H*) of the incandescence signals is used to infer whether the sampled aerosol contains the black carbon
(BC). If the *H* of the incandescence signal is larger than 500, the sample aerosol contains the BC and
the scattering signals should deviate from the signals of pure scattering aerosol. Those sample aerosols
are not considered when dealing with the aerosol scattering signals. This is achieved by just studying
the signals when the particle are recognized as pure scattering particle.
Despite that some aerosol particles are internally mixed with a small BC core, whose
incandescence signal is below the detection threshold of SP2, we will demonstrate that these particles
have little influence on the retrieved aerosol RRI. At the same time, there are some weakly absorbing
organic components that absorb light intensity in the near infrared range, which were termed as brown
carbon (BrC). These BrC components have ignorable influence on the retrieving of aerosol RRI, which
will be discussed in detail in section 4.2. Thus, the imaginary part of complex refractive index is set to
be zero in the following discussion.
**3 Methodology**
**3.1 Scattering intensity measured by the SP2**
From fig. 1(b), the APDs of the SP2 receive signals that were scattered by the sampled aerosols in
a certain small range at 45° and 135°. Thus, the scattering intensity ($S$) measured by the APD can be
expressed as:

$$S = C_0 \cdot I_0 \cdot \sigma \cdot (PF_{45^o} + PF_{135^o}) \qquad (6)$$

where $I_0$ is the laser's intensity; $\sigma$ is the scattering coefficient of the sampled aerosol, $PF_{45^o}$ and
$PF_{135^o}$ are scattering phase function at 45° and 135° respectively of the sampled aerosols; and $C_0$ is
a constant that is determined by the distance from the aerosol to the APD and the area of the APD. The
scattering intensity of the aerosol is recorded as the $H$ of the scattering signals in SP2. The following
calibration studies show that the scattering intensity S is highly related the H measured by SP2.
Therefore, the SP2 can be used as a powerful tool to measure the scattering signals of the sampled
aerosols, thus determining the corresponding scattering intensity.
Based on the Mie scattering theory, the scattering coefficient $\sigma$ can be calculated by integrating
the square of scattering intensity function $Q(\theta, x, RRI)$ from 0° to 180°. Angle $\theta$ is defined as the
angle between the light incident direction and scattering light direction. The size parameter x is defined
as $x = \frac{\pi D_p}{\lambda}$, where $\lambda$ is the light incident wavelength. The scattering phase function can be directly
derived from $S(\theta, x, m)$, too. Therefor, the $\sigma$, $PF_{45^o}$ and $PF_{135^o}$ in equation 6 are determined by
the $Dp$ and RRI of the aerosol. The amount of scattering signals from the sample aerosol varies with
the aerosol Dp and RRI (Bohren and Huffman, 2007). The scattering intensity at different aerosol
diameters and RRI is calculated based on equation 6 and shown in fig. 2. The $C_0$ is assumed to be 1
here. From fig. 2, we can see that the aerosol scattering intensity increases monotonously with the
increasing aerosol RRI at a given $Dp$, which makes it possible to retrieve the aerosol RRI with given
$Dp$ and the scattering intensity.
Bridging the scattering $H$ values measured by the SP2 scattering channel and the scattering
intensity S defined by equation 6 is achieved by calibrating the SP2 with ammonium sulfate. The
instrument setup of the calibration procedure is the same as that described in section 2.1. The diameters
of the aerosols passing through the DMA are manually changed from 100 to 450 nm with a step of 10
nm. For each diameter, the scattering H value and incandescence signal of every particle are analyzed.
When calibrating, there is no aerosol whose incandescence signal exceeds 1000 (This value depends
on the stability of the instrument and working conditions. It can be different for different instrument),
which means that the SP2 works stably and the incandescence signal channel can well distinguish the
BC containing aerosols. With the calibration, the relationship between the measured H and
theoretically calculated S can be determined.
The procedure of retrieving the RRI are summarized as follows: (1) measuring the scattering $H$
values at a given $Dp$; (2) transferring the $H$ into to $S$ by the established relationship from calibration;
(3) calculating the refractive index with the given $Dp$ and $S$ by using equation 6.
**3.2 Multiple Charging**
Fig. S1 gives the aerosols scattering H probability distribution under different aerosol diameters.
For each diameter, the distributions of the scattering H may have more than one mode for both the high
gain and low gain channels. The following discussions would give explanation about the multiple
mode distributions of $H$.
For each mode, the number of recorded aerosol particles at a given $H$ is fit by the log-normal
distribution function:
$$N(H) = \frac{N_0}{\sqrt{2\pi}\log(\sigma_g)} \cdot exp\left[-\frac{\log(H)-\log(H_0)}{2log^2(\sigma_g)}\right] \qquad (7)$$
Where $\sigma_g$ is the geometric standard deviation; $H_0$ is the geometric mean value of $H$ and $N_0$ is the
number concentrations for a peak mode. The geometric standard deviation is highly related to the half
width of the transfer function (equation 3). The $H_0$ is further used for discussion in the following part.
The $H_0$ values of corresponding to different elementary charges are labeled with different markers
in fig. 3. The $\sigma_g$ is fitted to be a small range at $1.182 \pm 0.02$ for different modes and different aerosol
diameters. In the following discussion, we conclude that the different $H_0$ values in fig. 3 represent that
the aerosols are charged with different number of elementary charges. Based on the Mie scattering
theory (Bohren and Huffman, 2007), the scattering intensity increases with increasing $Dp$, which imply
that the $H_0$ of the singly charged aerosol should increase with the increment of $Dp$. Thus, the black
square markers in fig. 3 represent the aerosols that are singly charged. At the same time, the
relationships between the $H_0$ and $Dp$ can be interpolated.

206        Other colored markers represent that the aerosols have more than one charge. We calculated the

corresponding diameter ($\widetilde{Dp}$) of the aerosols that share the same $Z_p$ but different charges with those
particles that have diameter of $Dp$ with one charge. Then the corresponding $\widetilde{H_0}$ at $\widetilde{Dp}$ are calculated.
Then the relationship between $\widetilde{H_0}$ and $Dp$ is shown in dashed line in fig. 3(a). From fig. 3(a), the
calculated $H_0$ shows good consistence with the measured $H_0$.

211        From the discussion above, we conclude that the SP2 can only detect those ammonium sulfate

aerosols with the diameter larger than 160 nm. However, the ambient aerosol RRI is always lower than
that of ammonium sulfate (Liu and Daum, 2008), thus the lower detecting limit of the ambient
scattering aerosols should be larger than 160 nm. The measured $H_0$ of the SP2 scattering low gain
channel signals are shown in fig. 3(b). From fig. 3(b), the same results can be deduced as those of the
high gain channel signals.

217        Fig. 4(a) gives the relationships between the calculated scattering intensity and the SP2 aerosol

scattering $H_0$ at different diameters. When calculating the scattering intensity, the RRI value of
ammonium sulfate is set to be 1.521 (Flores et al., 2009), and the $C_0$ in equation 6 is set to be unity.
The aerosol scattering intensity shows good consistence with the peak height ($R^2$=0.9992), which to
some extent reflect the high accuracy of our proposed method. When regressing the scattering intensity
on the measured peak height, the value 0.36 were obtained for the slope, which means that the
scattering intensity can be calculated by multiplying the peak height with a factor of 0.36.
**3.3 Validation of the calibration**

225        Ammonium chloride is used to validate the method of deriving the RRI from SP2. The RRI value

of ammonium chloride is 1.642 (Lide, 2006). The scattering H of the ammonium chloride under
different diameters are measured and analyzed. Fig. 4(b) shows the comparison between the measured
scattering high gain peak heights and the theoretical peak heights at different aerosols diameters.
Results show that the measured peak heights and the calculated ones are well correlated with
$R^2$=0.9994, which means that the DMA and SP2 can be used to derived the aerosol RRI with high
accuracy.
Fig. S2 gives the corresponding results of the scattering low gain channel. In fig. S2, the
relationship between the aerosol scattering peak height of the low gain channel and the scattering
intensity is determined. At the same time, the comparison between the measured peak height and the
calculated peak height shows good consistence too.
**4 Results and Discussion**
**4.1 Field Measurements**
Figure 5 shows the measured average probability distribution of the ambient size-resolved RRI
and the measured mean PNSD over two hours during the measurement. From fig. 5, we can see that
the derived RRI is $1.46 \pm 0.02$ and doesn't vary significantly with diameter between 199 nm and 436
nm. The aerosol chemical component may not vary significantly for different diameters during the
measurement. Another field measurement shows that the measured RRI varies significantly from 1.47
at 198 nm to 1.54 at 450 nm as shown in Fig. S4 in supplementary part.
The measured aerosol PNSD during the measurement has a maximum of 26400 #/cm$^3$ at 107 nm.
The mass concentration of the BC measured by the SP2 is 6.31 μg/m$^3$. Based on the measured PNSD
and the measured RRI, the size distribution of the scattering coefficient is calculated based on the Mie
scattering theory. The results in fig. 5 show that the measured RRI diameter range covers most of the
aerosol that contributes a fraction of 0.63 to the aerosol scattering properties with integrated scattering
coefficient at 385 Mm$^{-1}$. Thus, the derived size-resolved RRI of this range is representative of the
ambient aerosols scattering properties.
**4.2 Uncertainty analysis**
**4.2.1 Uncertainties from SP2**
The factors that influence the accuracy of retrieving RRI include the aerosols scattering $H$
measured by SP2 and the aerosol diameter selected by DMA.
The uncertainties of the selected diameter by DMA are well characterized based on equation 2 and
3. The uncertainties from the DMA transfer function can be avoided by fitting the scattering $H$ using
the log-normal distribution function. However, the uncertainties of the measured H from the SP2
remain unknown. The half width ($\Delta Z_p / Z_p$) of the transfer function is 0.1 times the scanning diameter,
which means that the geometric standard deviation of the aerosol PNSD selected by the DMA is
estimated to be 1.102. At the same time, the measured geometric standard deviation of the measured
*H* mode by SP2 is 1.182. Thus, the additional broadening by the H distribution is 1.073, which implies
that the geometric standard deviation of the measured *H* from the SP2 is estimated to be 1.073, whose
corresponding uncertainties is 6.8%.
The uncertainties of the retrieved RRI to the variations in the measured *H* are analyzed. Firstly,
we calculated the theoretical scattering intensity that can be measured by the SP2 for a given aerosol
diameter and RRI. The scattering intensity are changed by $\pm$6.8% and the corresponding RRI can be
derived using the given aerosol diameter and changed scattering intensity. Finally, the derived RRI are
compared with the given aerosol RRI. The uncertainties are analyzed for different aerosol diameter
and different RRI. The corresponding results are shown in fig. 6. The variations in RRI increase with
the increment of RRI but decrease with the increment of the *Dp*. For most ambient aerosols, the RRI
ranges from 1.4 to 1.5 and corresponds to a variation in RRI of 0.015.
Table 1 lists the retrieved ammonium chloride RRI under different diameters. The absolute
difference between the retrieved RRI and theoretical values is always smaller than 0.02 regardless of
the particle diameter, which means that the measured RRI is in line with the theoretical one. Thus, we
conclude that the uncertainty of the retrieved RRI is within 0.02 due to the uncertainties of SP2
measurement.
**4.2.2 Uncertainties due to BC exists**
There are some particles with a small soot core and the incandescence signal is below the detection
threshold of SP2. The derived aerosol RRI should be influenced by small soot core. Uncertainties
might be resulted when deriving the RRI for these BC-contained aerosols. With the calibration of the
SP2 with Aquadag soot particles, we concluded that the SP2 can't detect the soot particles lower than
80 nm, which is shown in detail in supplementary material in section S3.
We derived the aerosol equivalent refractive index when the aerosol have BC cores lower than 80
nm with two steps. The scattering strength of the BC-containing aerosols are first calculated based on
Mie scattering theory. Then the scattering strength are used to deriving the equivalent refractive index
with assuming that the BC-containing aerosols are pure scattering aerosols.
Monte Carlo simulations were applied to investigate the influence of the BC core on the retrieved
ambient aerosol RRI. Firstly, aerosol with diameter between 200 nm and 500 nm was chosen. Then
the core diameter are random determined lower than 80 nm. The core diameters flow the log-normal

distribution with the mean core diameter of 120 nm (Raatikainen et al., 2017). When calculating the scattering strength, the complex refractive index of the core 1.8+0.54$i$ (Zhao et al., 2018) is used. The complex refractive of the shell adopts the measured mean values (1.46+0$i$) during the field measurements. The scattering strength can be calculated with the above information. With the calculated scattering strength, the equivalent real part of the refractive index (RRI) can be derived with assuming that the aerosols are pure scattering aerosols. If the core diameter is 0, then the derived aerosol equivalent aerosol RRI should be 1.46.

For each aerosol diameter, the Monte Carlo simulations were conducted for 10000 times. Fig. 7(a) gives the retrieved aerosol equivalent RRI at different diameters. Results show that the retrieved aerosol equivalent RRI are larger than 1.46 for all of the given aerosol diameters. When the aerosols have BC core, the scattering strength are larger than that of pure scattering aerosols with the same aerosol diameter. The derived mean equivalent RRI tend to be closer to 1.46 when the aerosol diameter are larger, where the BC core contributes less and the influence of the BC core are be smaller. The derived mean aerosol equivalent RRI is 1.47 and 1.462 at 200 nm and 500 nm respectively. At the same time, the uncertainties associated with the equivalent RRI are larger when the aerosol diameter are smaller. We conclude that the uncertainties associated with BC core are smaller than 0.01 when the aerosol diameter are larger than 250 nm. The maximum of the difference of the derived RRI is 0.02.

### 4.2.3 Uncertainties from BrC

There are some BrCs that absorb the light intensity in the near infrared range. The imaginary part of the refractive index at a given wavelength $\lambda$ ($k_\lambda$) of the BrC can be calculated as:

$$k_{\lambda 1} = k_{\lambda 2} \times (\frac{\lambda 2}{\lambda 1})^w \qquad (8)$$

Where $w$ is defined by mass of BC to organic aerosol ratio (R) (Saleh et al., 2015)with:

$$w = \frac{0.21}{R+0.07} \qquad (9)$$

Based on the work of Saleh et al. (2015), the $k_{550}$ can be expressed as:

$$k_{550} = 0.016 \times log_{10}(R) + 0.04 \qquad (10)$$

The values R ranges between 0.09 and 0.35 for different types of aerosols (Saleh et al., 2015). Based on equation (8), (9) and (10), the $k_{1024}$ ranges between 0.01 and 0.024. The maximum value 0.024 is used for further analysis.

The uncertainties of the retrieved RRI when ignoring the effect of BrC are analyzed. Firstly, The scattering light intensity at a given diameter with a refractive index of $1.46 + 0.024i$ is calculated using the Mie model. Then the corresponding RRI are retrieved using given diameter and the calculated light intensity with assumption that these are pure scattering aerosols. The retrieved aerosol RRI values for different aerosol diameter are shown in fig. 7(b). For the light absorbing particles, their scattering light intensity is smaller than that of the pure scattering particles with the same diameter and RRI. Therefore, the retrieved aerosol RRI is underestimated for most of the conditions. The differences between the given RRI value (1.46) and retrieved RRI value are lower than 0.006 for all of the diameters as shown in fig. 7(b). The BrC component has little influence on the retrieved aerosol RRI.

### 4.2.4 Overall of the uncertainties

Monte Carlo simulations were conducted to study the influence of the above three uncertainty sources. Four steps are involved in the Monte Carlo simulations. First, the core diameter of an aerosol particle at a given diameter are randomly given with the core diameter flowing the log-normal distribution with the mean core diameter of 120 nm (Raatikainen et al., 2017). The refractive index of the core is set to be the same as that in section 4.2.2. The RRI of the shell uses the measured mean value 1.46. The imaginary part of the shell is determined randomly with a mean value of 0.023. Second, the light scattering intensity can be calculated using the Mie model and the information in step one. Then the light scattering intensity was randomly changed with uncertainties of 6.8%. Finally, the changed light scattering intensity are used to derive the aerosol RRI with the given diameter and assumption that the particles are pure scattering particles.

The aerosol diameters were changed from 200 nm to 500 nm, and the simulations were conducted for 10000 times for each diameter. The overall uncertainties are shown in fig. 7(c). The uncertainties from SP2 instrument measurement don't lead to bias of the retrieved aerosol RRI. When the aerosol diameter is lower than 300 nm, the influence of the BC core is more important than the influence of BrC. The retrieved RRI tend to be overestimated when the aerosol is lower than 300 nm. When the aerosol diameter is larger than 300 nm, the influence of BrC domains and the retrieved aerosol RRI are underestimated. However, the bias caused by BC and BrC are all the way lower than 0.01. For most of the conditions, the retrieved aerosol RRI are within the range of $1.46 \pm 0.02$. Thus, we conclude that the uncertainty of the retrieved RRI is 0.02 with considering all of the factors.

**5 Conclusions**

Knowledge on the microphysical properties of ambient aerosol is import for better evaluating their radiative forcing. The aerosol RRI is a key factor that determines the aerosol scattering properties. In this study, a new method to measure the ambient aerosol RRI is developed by synthetically using a DMA in tandem with a SP2. This method can continuously measure the size-resolved RRI over a wide range between 198 nm and 426 nm. At the same time, it is free from the influence of the BC containing aerosols.

The basic principle of measuring the size-resolved RRI is to select the aerosols at a certain diameter by the DMA and measure the corresponding scattering intensity by the SP2. The relationship between the aerosols scattering intensity and the peak height of the scattering signal channels are determined by calibrating the SP2 using ammonium sulfate (RRI=1.521).

The method is validated by measuring the size-resolved RRI of the ammonium chloride with the RRI value of 1.642 as sample aerosol and the corresponding derived value is $1.642 \pm 0.02$. There are three factors that influence the accuracy of derived aerosol RRI. The measured scattering intensity by SP2 has an uncertainty of 0.68%, which can lead to the uncertainties of the derived RRI values less than 0.15. There are some particles with a small soot core and the incandescence signal is below the detection threshold of SP2. The light scattering intensity of these particles increases compared with that of the pure scattering particles with the same aerosol diameters. The retrieved aerosol RRI values can be overestimated by up to 0.02. Some BrCs absorb the light intensity in the near infrared range. The corresponding scattering intensity is weaker than that of pure scatter particles for the same diameter and the retrieved aerosol RRI value can be underestimated by up to 0.006. Based on Monte Carlo simulations, the uncertainty of the retrieved RRI is 0.02 with considering all of the factors.

This instrument is employed at a field measurement at the AERONET PKU stating, the size-resolved RRI of the ambient aerosols is 1.46 and doesn't show significant variation among the diameter. The corresponding aerosol diameter range, which can be detected by SP2 to derive the RRI, covers most of the aerosol scattering. Thus, the derived size-resolved RRI of this range can be used as a good representative of the ambient aerosols scattering properties.

*Data availability.* The measurement data involved in this study are available upon request to the authors.

*Author contributions.* Gang Zhao and Chunsheng Zhao designed the experiments; Gang Zhao and Weilun Zhao conducted the measurements; Chunsheng Zhao and Gang Zhao discussed the results and wrote the manuscript.

*Competing interests.* The authors declare that they have no conflict of interest.

*Acknowledgments.* This work is supported by the National Key R&D Programa of China (2016YFC020000:Task 5) and the National Natural Science Foundation of China (41590872).

Barkey, B., Paulson, S. E., and Chung, A.: Genetic Algorithm Inversion of Dual Polarization Polar Nephelometer Data to Determine Aerosol Refractive Index, Aerosol Sci. Technol., 41, 751-760, 10.1080/02786820701432640, 2007.

Bohren, C. F., and Huffman, D. R.: Absorption and Scattering by a Sphere, in: Absorption and Scattering of Light by Small Particles, Wiley-VCH Verlag GmbH, 82-129, 2007.

Cai, Y., Montague, D. C., and Deshler, T.: Comparison of measured and calculated scattering from surface aerosols with an average, a size-dependent, and a time-dependent refractive index, Journal of Geophysical Research, 116, 10.1029/2010jd014607, 2011.

Dick, W. D., Ziemann, P. J., and McMurry, P. H.: Multiangle Light-Scattering Measurements of Refractive Index of Submicron Atmospheric Particles, Aerosol Sci. Technol., 41, 549-569, 10.1080/02786820701272012, 2007.

Dubovik, O.: Variability of absorption and optical properties of key aerosol types observed in worldwide locations, J.atmos.sci, 59, 590-608, 2002.

Ebert, M., Weinbruch, S., Rausch, A., Gorzawski, G., Helas, G., Hoffmann, P., and Wex, H.: Complex refractive index of aerosols during LACE 98#x2010; as derived from the analysis of individual particles, Journal of Geophysical Research: Atmospheres, 107, LAC 3-1-LAC 3-15, 10.1029/2000jd000195, 2002.

Ebert, M., Weinbruch, S., Hoffmann, P., and Ortner, H. M.: The chemical composition and complex refractive index of rural and urban influenced aerosols determined by individual particle analysis, Atmospheric Environment, 38, 6531-6545, 10.1016/j.atmosenv.2004.08.048, 2004.

Eidhammer, T., Montague, D. C., and Deshler, T.: Determination of index of refraction and size of supermicrometer particles from light scattering measurements at two angles, Journal of Geophysical Research, 113, 10.1029/2007jd009607, 2008.

Flores, J. M., Trainic, M., Borrmann, S., and Rudich, Y.: Effective broadband refractive index retrieval by a white light optical particle counter, Phys Chem Chem Phys, 11, 7943-7950, 10.1039/b905292e, 2009.

Ghan, S. J., and Schwartz, S. E.: Aerosol Properties and Processes: A Path from Field and Laboratory Measurements to Global Climate Models, Bulletin of the American Meteorological Society, 88, 1059-1084, 10.1175/bams-88-7-1059, 2007.

Hand, J. L., and Kreidenweis, S. M.: A New Method for Retrieving Particle Refractive Index and Effective Density from Aerosol Size Distribution Data, Aerosol Sci. Technol., 36, 1012-1026, 10.1080/02786820290092276, 2002.

Hänel, G.: REAL PART OF MEAN COMPLEX REFRACTIVE INDEX AND MEAN DENSITY OF SAMPLES OF ATMOSPHERIC AEROSOL PARTICLES, Tellus, 20, 371-&, 10.3402/tellusa.v20i3.10016, 1968.

Hu, M., Peng, J., Sun, K., Yue, D., Guo, S., Wiedensohler, A., and Wu, Z.: Estimation of size-resolved ambient particle density based on the measurement of aerosol number, mass, and chemical size distributions in the winter in Beijing, Environ Sci Technol, 46, 9941-9947, 10.1021/es204073t, 2012.

Kandler, K., Benker, N., Bundke, U., Cuevas, E., Ebert, M., Knippertz, P., Rodríguez, S., Schütz, L., and Weinbruch, S.: Chemical composition and complex refractive index of Saharan Mineral Dust at Izaña, Tenerife (Spain) derived by electron microscopy, Atmospheric Environment, 41, 8058-8074, 10.1016/j.atmosenv.2007.06.047, 2007.

Knutson, E. O., and Whitby, K. T.: Aerosol classification by electric mobility: apparatus, theory, and applications, Journal of Aerosol Science, 6, 443-451, https://doi.org/10.1016/0021-8502(75)90060-9, 1975.

Kuang, Y., Zhao, C. S., Tao, J. C., and Ma, N.: Diurnal variations of aerosol optical properties in the North China Plain and their influences on the estimates of direct aerosol radiative effect, Atmos. Chem.

Phys., 15, 5761-5772, 10.5194/acp-15-5761-2015, 2015.
Levoni, C., Cervino, M., Guzzi, R., and Torricella, F.: Atmospheric aerosol optical properties: a
database of radiative characteristics for different components and classes, Appl Opt, 36, 8031-8041,

439     1997.

Lide, D. R.: Handbook of Chemistry and Physics, 86th Edition Edited(National Institute of Standards
and Technology), Journal of the American Chemical Society, 128, 5585-5585, 10.1021/ja059868l,

442     2006.

Liu, H. J., Zhao, C. S., Nekat, B., Ma, N., Wiedensohler, A., van Pinxteren, D., Spindler, G., Müller,
K., and Herrmann, H.: Aerosol hygroscopicity derived from size-segregated chemical composition and
its parameterization in the North China Plain, Atmospheric Chemistry and Physics, 14, 2525-2539,
10.5194/acp-14-2525-2014, 2014.
Liu, Y., and Daum, P. H.: THE EFFECT OF REFRACTIVE INDEX ON SIZE DISTRIBUTIONS
AND LIGHT SCATTERING COEFFICIENTS DERIVED FROM OPTICAL PARTICLE
COUNTERS ☆, Journal of Aerosol Science, 31, 945-957, 2000.
Liu, Y., and Daum, P. H.: Relationship of refractive index to mass density and self-consistency of
mixing rules for multicomponent mixtures like ambient aerosols, Journal of Aerosol Science, 39, 974-
986, 10.1016/j.jaerosci.2008.06.006, 2008.
Moffet, R. C., Qin, X., Rebotier, T., Furutani, H., and Prather, K. A.: Chemically segregated optical
and microphysical properties of ambient aerosols measured in a single-particle mass spectrometer,
Journal of Geophysical Research, 113, 10.1029/2007jd009393, 2008.
Moise, T., Flores, J. M., and Rudich, Y.: Optical properties of secondary organic aerosols and their
changes by chemical processes, Chemical Reviews, 115, 4400-4439, 2015.
Peng, J., Hu, M., Guo, S., Du, Z., Zheng, J., Shang, D., Levy Zamora, M., Zeng, L., Shao, M., Wu, Y.-
S., Zheng, J., Wang, Y., Glen, C. R., Collins, D. R., Molina, M. J., and Zhang, R.: Markedly enhanced
absorption and direct radiative forcing of black carbon under polluted urban environments,
Proceedings of the National Academy of Sciences, 113, 4266-4271, 10.1073/pnas.1602310113, 2016.
Qiao, K., Wu, Z., Pei, X., Liu, Q., Shang, D., Zheng, J., Du, Z., Zhu, W., Wu, Y., Lou, S., Guo, S.,
Chan, C. K., Pathak, R. K., Hallquist, M., and Hu, M.: Size-resolved effective density of submicron
particles during summertime in the rural atmosphere of Beijing, China, Journal of Environmental
Sciences, 10.1016/j.jes.2018.01.012, 2018.
Raatikainen, T., Brus, D., Hooda, R. K., Hyvärinen, A.-P., Asmi, E., Sharma, V. P., Arola, A., and
Lihavainen, H.: Size-selected black carbon mass distributions and mixing state in polluted and clean
environments of northern India, Atmospheric Chemistry and Physics, 17, 371-383, 10.5194/acp-17-

469    371-2017, 2017.

Ramanathan, V., and Carmichael, G.: Global and regional climate changes due to black carbon, Nature
Geoscience, 1, 221-227, 10.1038/ngeo156, 2008.
Redemann, J., Turco, R. P., Liou, K. N., Russell, P. B., Bergstrom, R. W., Schmid, B., Livingston, J.
M., Hobbs, P. V., Hartley, W. S., Ismail, S., Ferrare, R. A., and Browell, E. V.: Retrieving the vertical
structure of the effective aerosol complex index of refraction from a combination of aerosol in situ and
remote sensing measurements during TARFOX, Journal of Geophysical Research: Atmospheres, 105,
9949-9970, 10.1029/1999jd901044, 2000.
Saleh, R., Marks, M., Heo, J., Adams, P. J., Donahue, N. M., and Robinson, A. L.: Contribution of
brown carbon and lensing to the direct radiative effect of carbonaceous aerosols from biomass and
biofuel burning emissions, Journal of Geophysical Research: Atmospheres, 120, 10,285-210,296,
doi:10.1002/2015JD023697, 2015.
Schwarz, J. P., Gao, R. S., Fahey, D. W., Thomson, D. S., Watts, L. A., Wilson, J. C., Reeves, J. M.,
Darbeheshti, M., Baumgardner, D. G., Kok, G. L., Chung, S. H., Schulz, M., Hendricks, J., Lauer, A.,
Kärcher, B., Slowik, J. G., Rosenlof, K. H., Thompson, T. L., Langford, A. O., Loewenstein, M., and
Aikin, K. C.: Single-particle measurements of midlatitude black carbon and light-scattering aerosols
from the boundary layer to the lower stratosphere, Journal of Geophysical Research, 111,
10.1029/2006jd007076, 2006.
Shepherd, R. H., King, M. D., Marks, A. A., Brough, N., and Ward, A. D.: Determination of the
refractive index of insoluble organic extracts from atmospheric aerosol over the visible
wavelength range using optical tweezers, Atmospheric Chemistry and Physics, 18, 5235-5252,
10.5194/acp-18-5235-2018, 2018.
Sorooshian, A., Hersey, S., Brechtel, F. J., Corless, A., Flagan, R. C., and Seinfeld, J. H.: Rapid, Size-
Resolved Aerosol Hygroscopic Growth Measurements: Differential Aerosol Sizing and
Hygroscopicity Spectrometer Probe (DASH-SP), Aerosol Sci. Technol., 42, 445-464,

494    10.1080/02786820802178506, 2008.

Spindler, C., Riziq, A. A., and Rudich, Y.: Retrieval of Aerosol Complex Refractive Index by
Combining Cavity Ring Down Aerosol Spectrometer Measurements with Full Size Distribution
Information, Aerosol Sci. Technol., 41, 1011-1017, 10.1080/02786820701682087, 2007.
Stelson, A. W.: Urban aerosol refractive index prediction by partial molar refraction approach,
Environ.sci.technol, 24:11, 1676-1679, 1990.
Stephens, M., Turner, N., and Sandberg, J.: Particle identification by laser-induced incandescence in a
solid-state laser cavity, Appl Opt, 42, 3726-3736, 2003.
Valenzuela, A., Reid, J. P., Bzdek, B. R., and Orr-Ewing, A. J.: Accuracy required in measurements of
refractive index and hygroscopic response to reduce uncertainties in estimates of aerosol radiative
forcing efficiency, Journal of Geophysical Research: Atmospheres, 10.1029/2018jd028365, 2018.
Vratolis, S., Fetfatzis, P., Argyrouli, A., Papayannis, A., Müller, D., Veselovskii, I., Bougiatioti, A.,
Nenes, A., Remoundaki, E., Diapouli, E., Manousakas, M., Mylonaki, M., and Eleftheriadis, K.: A
new method to retrieve the real part of the equivalent refractive index of atmospheric aerosols, Journal
of Aerosol Science, 117, 54-62, 10.1016/j.jaerosci.2017.12.013, 2018.
Wang, Y., Wu, Z., Ma, N., Wu, Y., Zeng, L., Zhao, C., and Wiedensohler, A.: Statistical analysis and
parameterization of the hygroscopic growth of the sub-micrometer urban background aerosol in
Beijing, Atmospheric Environment, 10.1016/j.atmosenv.2017.12.003, 2017.
Wendisch, M., and Hoyningen-Huene, W. V.: Optically equivalent refractive index of atmospheric
aerosol particles, Huene, 65, 1992.
Wendisch, M., and Hoyningen-Huene, W. V.: Possibility of refractive index determination of
atmospheric aerosol particles by ground-based solar extinction and scattering measurements,
Atmospheric Environment, 28, 785-792, 1994.
Wex, H., Neusüß, C., Wendisch, M., Stratmann, F., Koziar, C., Keil, A., Wiedensohler, A., and Ebert,
M.: Particle scattering, backscattering, and absorption coefficients: An in situ closure and sensitivity
study, Journal of Geophysical Research: Atmospheres, 107, LAC 4-1-LAC 4-18,
10.1029/2000jd000234, 2002.
Wiedensohler, A., Birmili, W., Nowak, A., Sonntag, A., Weinhold, K., Merkel, M., Wehner, B., Tuch,
T., Pfeifer, S., Fiebig, M., Fjäraa, A. M., Asmi, E., Sellegri, K., Depuy, R., Venzac, H., Villani, P., Laj,
P., Aalto, P., Ogren, J. A., Swietlicki, E., Williams, P., Roldin, P., Quincey, P., Hüglin, C., Fierz-
Schmidhauser, R., Gysel, M., Weingartner, E., Riccobono, F., Santos, S., Grüning, C., Faloon, K.,

Beddows, D., Harrison, R., Monahan, C., Jennings, S. G., O'Dowd, C. D., Marinoni, A., Horn, H. G., Keck, L., Jiang, J., Scheckman, J., McMurry, P. H., Deng, Z., Zhao, C. S., Moerman, M., Henzing, B., de Leeuw, G., Löschau, G., and Bastian, S.: Mobility particle size spectrometers: harmonization of technical standards and data structure to facilitate high quality long-term observations of atmospheric particle number size distributions, Atmospheric Measurement Techniques, 5, 657-685, 10.5194/amt-5-657-2012, 2012.

Zarzana, K. J., Cappa, C. D., and Tolbert, M. A.: Sensitivity of Aerosol Refractive Index Retrievals Using Optical Spectroscopy, Aerosol Sci. Technol., 48, 1133-1144, 10.1080/02786826.2014.963498, 2014.

Zhang, G., Bi, X., Han, B., Qiu, N., Dai, S., Wang, X., Sheng, G., and Fu, J.: Measurement of aerosol effective density by single particle mass spectrometry, Science China Earth Sciences, 59, 320-327, 10.1007/s11430-015-5146-y, 2015.

Zhang, Y., Zhang, Q., Cheng, Y., Su, H., Kecorius, S., Wang, Z., Wu, Z., Hu, M., Zhu, T., Wiedensohler, A., and He, K.: Measuring the morphology and density of internally mixed black carbon with SP2 and VTDMA: new insight into the absorption enhancement of black carbon in the atmosphere, Atmospheric Measurement Techniques, 9, 1833-1843, 10.5194/amt-9-1833-2016, 2016.

Zhao, G., Zhao, C., Kuang, Y., Tao, J., Tan, W., Bian, Y., Li, J., and Li, C.: Impact of aerosol hygroscopic growth on retrieving aerosol extinction coefficient profiles from elastic-backscatter lidar signals, Atmospheric Chemistry and Physics, 17, 12133-12143, 10.5194/acp-17-12133-2017, 2017.

Zhao, G., Zhao, C., Kuang, Y., Bian, Y., Tao, J., Shen, C., and Yu, Y.: Calculating the aerosol asymmetry factor based on measurements from the humidified nephelometer system, Atmospheric Chemistry and Physics, 18, 9049-9060, 10.5194/acp-18-9049-2018, 2018.


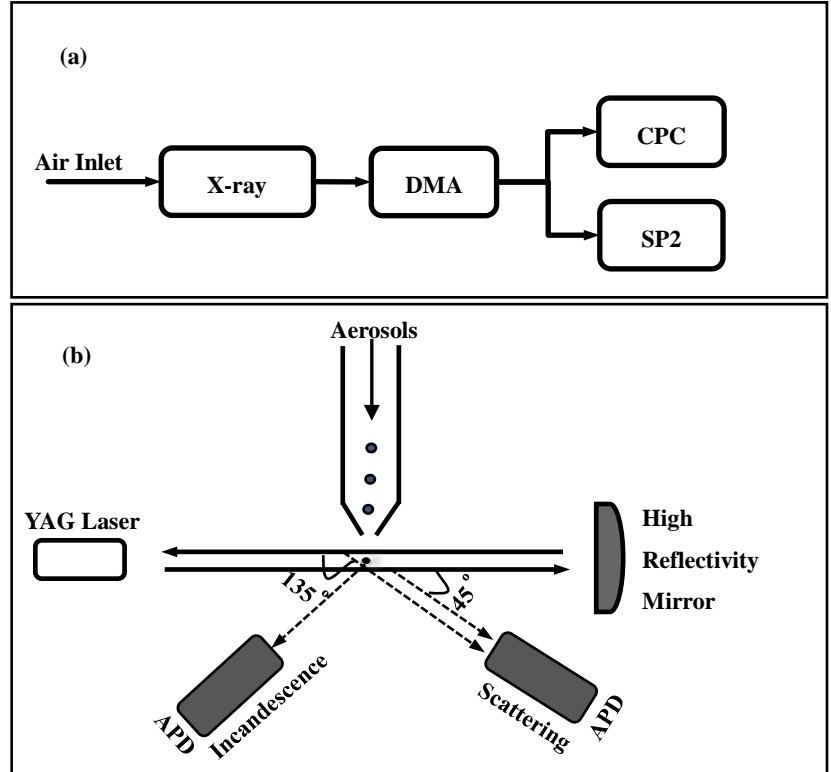

**Figure 1.** (a) Schematic of the measurement system. (b) Diagram of SP2 Chamber.


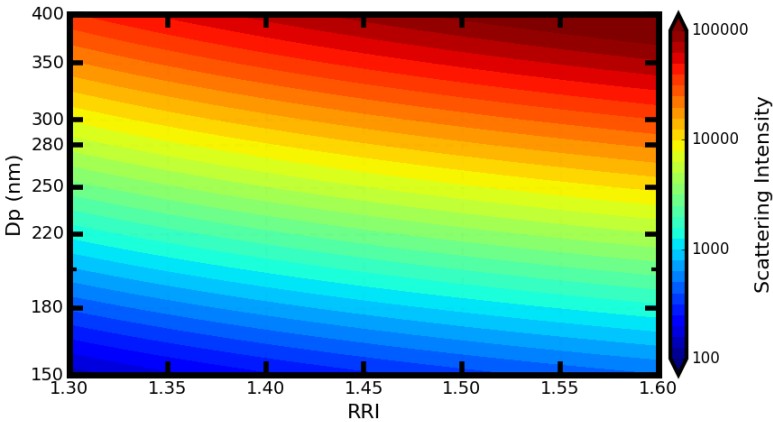

**Figure 2.** The distribution of the aerosols scattering intensity at different Dp and different RRI.

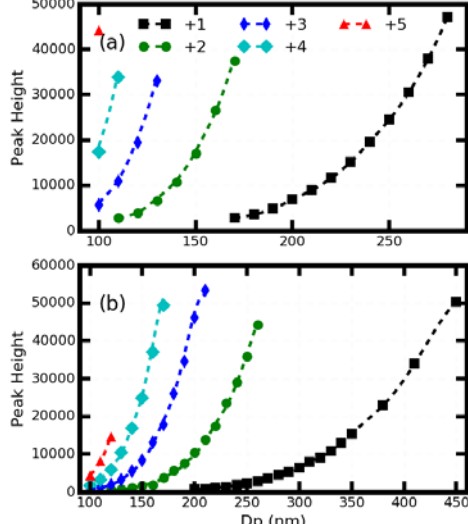

**Figure 3.** The geometric mean peak height for different diameters of the high gain channel. The markers gives the measured values and the dotted line shows the theoretically calculated value. Different colors represent the different number of elementary charges.

558

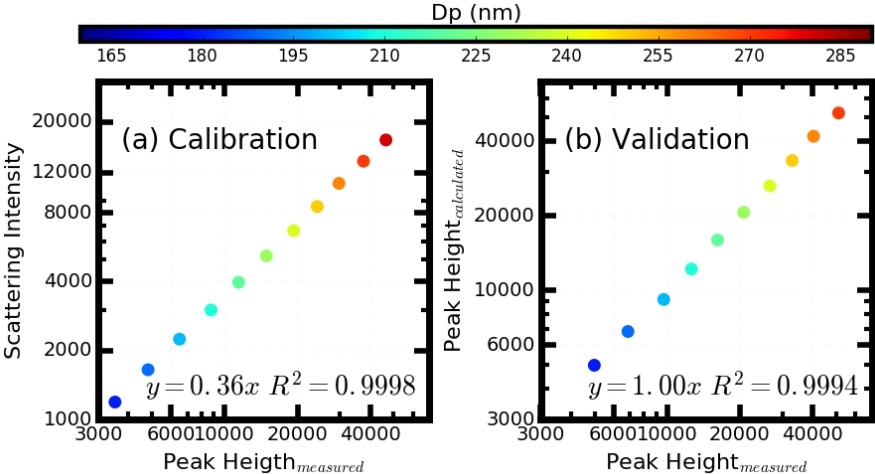

**Figure 4.** (a) the relationship between the scattering peak height from the SP2 high gain scattering channel when calibrating by using the ammonia sulfate and (b) the comparison between the measured scattering peak height from SP2 high gain scattering channel using the ammonia chloride and the calculated scattering peak height using the Mie scattering theory. Different colors represents the results at different diameters.


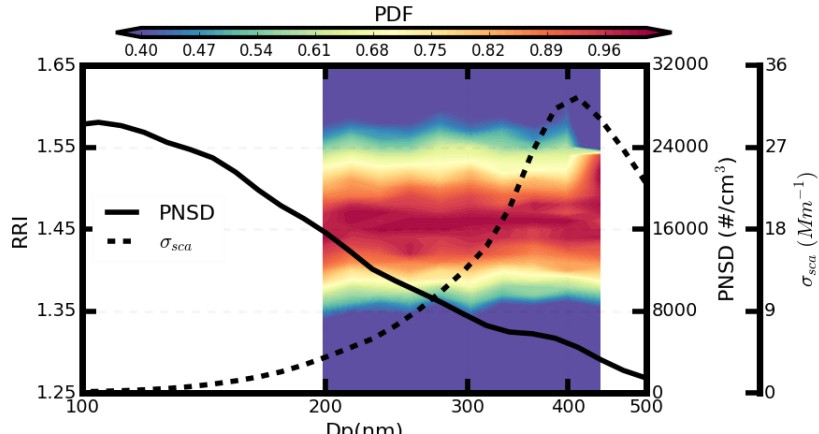

**Figure 5.** The measured probability of the size-resolved RRI (the filled color), the measured mean
PNSD (the full line) and the mean scattering size distribution (the dotted line).

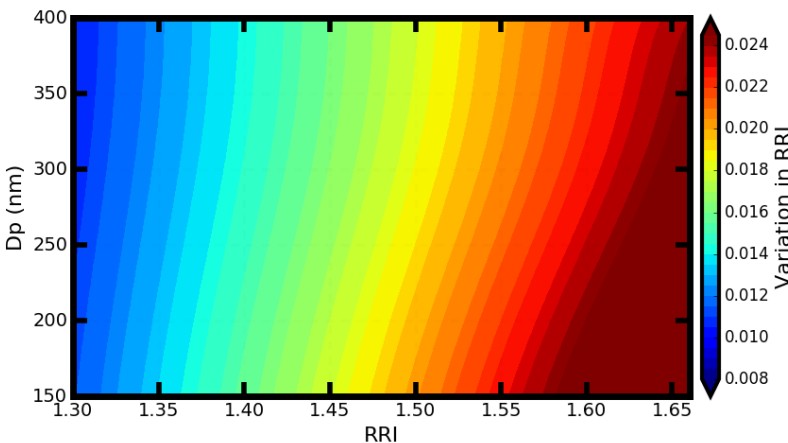

**Figure 6.** The variation in RRI for different kinds of aerosols that have different diameters and different RRI.

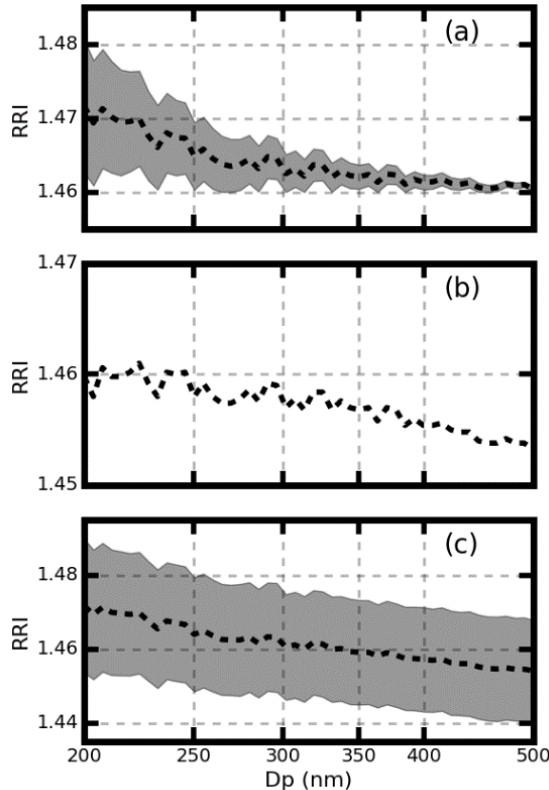


**Figure 7.** The retrieved aerosol RRI at different aerosol diameter. The filled color represents the 5[th]
and 95[th] percentiles.



**Tabel 1.** The retrieved RRI and the absolute difference between the retrieved RRI and the theoretical
RRI for different ammonia chloride diameters.

| Dp(nm) | 160 | 170 | 180 | 190 | 200 | 210 | 220 | 230 | 240 | 250 | 260 | 270 |
|---|---|---|---|---|---|---|---|---|---|---|---|---|
| RRI | 1.654 | 1.650 | 1.651 | 1.643 | 1.656 | 1.645 | 1.633 | 1.626 | 1.634 | 1.626 | 1.624 | 1.625 |
| Difference | 0.012 | 0.008 | 0.009 | 0.001 | 0.012 | 0.003 | 0.009 | 0.016 | 0.008 | 0.016 | 0.018 | 0.017 |
