# Peer review of "Method to measure the size-resolved real part of aerosol refractive index using"

_Atmospheric Measurement Techniques, 2018_

## Referee Comment (RC1) · Anonymous Referee #1 · 13 Jan 2019

In this manuscript, the authors introduced a method to retrieve the real part of refractive index (RRI) of ambient aerosols from the measurements of the scattering intensities of size-selected aerosol particles by the combination of the differential mobility analyzer (DMA) and the single particle soot photometer (SP2). The authors pointed out that retrieval of the size-resolved RRI of ambient aerosols is the innovation of this paper in comparison with the measurements of the total equivalent aerosol RRI or aerosol RRI at a given diameter in previous studies. It is a meaningful topic to measure the real part of the complex refractive index of ambient aerosol particles. However, there are still some important issues should be considered before it is publishable.

1. The authors pointed out that "there might be significant variations in the aerosol RRI for aerosols of different diameter because the aerosol RRI is highly related to the

aerosol density and chemical components...information of the size-resolved aerosol RRI can help to study the chemical information and the aging process of aerosols among different diameters". However, the results of the size-resolved RRI of the ambient aerosols do not show significant variations among different diameters. The authors should give explanations.

2. The size-resolved aerosol RRI is retrieved based on the Mie scattering theory at a given particle diameter. What is the effect of the imaginary part of the complex refractive index on the retrieval?

3. The impact of non-sphericity of ambient aerosols on the light scattering cannot be neglected, especially for dry particles. The authors should also discuss the uncertainties introduced by the sphericity assumption based on the Mie theory.

More specific comments:

1. Some details of the method to retrieval real part of the refractive index based on the Mie scattering theory should be added.

2. Lines 57-58: the authors pointed out that "Up to now, there is no information in the literature of the size-resolved ambient aerosol RRI over the diameter range between 200nm and 500nm...". However, the individual particle analysis combining scanning and transmission electron microscopy (SEM and TEM) have been widely used to derive size-resolved information of the complex refractive index of atmospheric aerosol particles (e.g., in the size range from 100 nm up to 50 $\mu$m in diameter) (Ebert et al.,2002, 2004; Kandler et al., 2007).

Ebert, M., et al., 2002. Complex refractive index of aerosols during LACE 98 as derived from the analysis of individual particles. Journal of Geophysical Research 107 (D21), 8121.

Ebert, M., Weinbruch, S., Hoffmann, P., Ortner, H.M., 2004.The chemical composition and complex refractive index ofrural and urban influenced aerosols determined by

individual particle analysis. Atmospheric Environment 38, 6531–6545.

Kandler K, Benker N, Bundke U, et al. 2007. Chemical composition and complex refractive index of Saharan Mineral Dust at Izaña, Tenerife (Spain) derived by electron microscopy. Atmospheric Environment 41(37), 8058-8074.

3. Section 4.1: The field measurements were carried out at the AERONET BEI-JING_PKU station. The results should be compared with the AERONET retrievals considering that the size-resolved RRI of the ambient aerosols doesn't show significant variation among different diameters.

4. Lines 232-233: "For most ambient aerosols, the RRI ranges from 1.4 to 1.5 …". Some researches have reported the values around 1.53∼1.57 for the RRI of most of dry components of atmospheric aerosols, and higher values for RRI of black carbon (BC) component (Xie et al., 2017). The authors should demonstrate their results with other measurements.

Xie Y S, Li Z Q, Zhang Y X, et al.,2017. Estimation of atmospheric aerosol composition from ground-based remote sensing measurements of Sun-sky radiometer. Journal of Geophysical Research Atmospheres122(1),498-518.

Typos/Grammar:

1. Line 12 and some other lines in the text: a space should be placed between the number and the unit.

2. Lines 27-29 and some other lines in the text: it is better to use the past tense in review of the literature.

3. Line 29: please rewrite the sentence "Valenzuela et al. (2018) also reports an uncertainty of 7% with the uncertainties of RRI of 0.1 in RRI."

4. Line 58: "the diameter range between 200nm and 500nm where the aerosol scattering coefficients contributes to…". "contributes" should be "contribute".

5. Line 64 and some other lines in the text: "for aerosol of different diameter" should be "for aerosol of different diameters"

6. Line 90: "PNSD" first appears in Section 2.1, but it has not been defined.

7. Lines 99-100 and some other places in the text: the physical quantities "V" and "Zp" should be set in italic in consistent with the equation.

8. Line 102: "L" in Eq. (2) has not been defined.

9. Lines 113 and 140: please distinguish the two "C" in Eqs. (5) and (6).

10. Lines 150 and 155: "equation (6)" and "equation 6" should be in a uniform format.

11. Line 156: "as that described in section 2.2.1". There is no section 2.2.1 in the manuscript.

12. Line 177: "PH0" first appears in Section 3.2, but it has not been defined.

13. Line 184: "Dp∼ " first appears in Section 3.2, but it has not been defined.

14. Lines 175, 180, 184-185: "fig.2" should be changed into "fig.3".

15. Line 221: "SP" should be "SP2".

16. Line 251: "This instrument is employed at a field measurement at the AERONET PKU stating. . .", please rewrite this sentence.

Please also note the supplement to this comment:
https://www.atmos-meas-tech-discuss.net/amt-2018-399/amt-2018-399-RC1-supplement.pdf

---

## Referee Comment (RC2) · Anonymous Referee #2 · 10 Apr 2019

General comments:

The present manuscript describes a method for deriving the real part of the refractive index by means of a differential mobility selector (DMA) and scattering intensities measured with a SP2. The derivation of the real part of the refractive index of a quasi mono disperse aerosol is not completely new. What is new, however, is the application with the use of the SP2, which in a unique way can also determine the mixing state of the aerosol within certain limits. This ensures that errors caused by unknown imaginary parts of the refractive index are minimised. The method shown is limited to non-absorbent particles.

The reviewer thinks that the current limitations and consequences have not been adequately presented. In particular, a consideration of the uncertainties in violation of the

restrictions (weakly absorbing particles) is lacking.

In laboratory experiments, as shown with ammonium sulfate and ammonium chloride, this is easily possible. The application to a complex ambient aerosol, on the other hand, was not treated sufficiently. The example shown in chapter 4.1 shows results of measurements in Beijing, where a complex mixed aerosol is present (that measurement place was characterised as urban roadside; line 96).

An error estimation is missing: a) what happens with weakly absorbing organic droplets, b) what happens with internally mixing particles with a small soot core when the incandescence signal is below the detection threshold of SP2. How large are the expected errors in the real part of the refractive index?

The reviewer believes that this work can make a good step in the optical characterisation of sub-micrometer particles using the SP2, and that subsequent work can build on it. Therefore, the reviewer thinks that the manuscript can be published after a major revision.

Specific comments:

Title: The method shown is very general, but applied to SP2 in the present study. The study is thus adapted to the size range, the size resolution and the optical geometry of the SP2. The author should consider whether the application of the method to SP2 should be mentioned in the title.

Introduction: Can the author give a first estimate on the accuracy of RRI measurements required to make statements on the chemical composition?

Line 36: Typo: "Hänel"

Lines 89, 90: The measurements provide the necessary data in five minute intervals. However, no conclusion can yet be drawn that the RRI can be derived with a time resolution of five minutes.

Line 121: to be precise, the power is about 1 W/m2 circulating power in cavity

Line 131: what is the unit of the peak height H.

Lines 133,134: How were BC containing particles ruled out for ambient measurements.

Section 3.1: Shouldn't the signals be the same value at 45° and 135° due to the circulating wave in the cavity?

Line 145: To avoid misunderstandings: The SP2 can determine the scattering signal in a certain scattering angle range. But not the scattering coefficient!

Figure 2: Is the scattering strength the same as the scattering intensity S? Please use consistent notations.

Line 151: monotonously instead of homogeneously

Line 159: Establishing the threshold value at 1000 seems somewhat arbitrary. Is there a justification for this?

Line 165: It would be better to read to bring figure S2 und figure 3 together.

Lines 166 and 167: Refer to figure S1

Line 175: Please check that sentence. I can't see different marker for different diameters. Is fig. 2 the correct figure?

Line 177: What is "PH0"?

Line 180: Shouldn't it be Fig 3.

Lines 180,181 and Figure 3: Check if the peak height is plotted versus the mobility diameter Zp and not versus the geometric diameter Dp?

Line 184: "Dp superscript tilde" not defined

Line 184: There is no dashed line in figure 2

Lines 194 and 200 : Please bring references for the refractive index

Figure 4a: Should be scattering intensity instead of scattering strength

Lines 192 – 208, Figure 4: For the reader it is not obvious at first sight which value was calibrated! What is the value of the calibration factor C? The reviewer thinks it is worth giving a short summary list of all steps necessary for deriving RRI. For an absolute calibration, the slope of about unity is more important than the references to the correlation coefficient. The high correlation coefficient is, as written, a good indicator for the potentially high accuracy of this method.

Line 211: It would be good to have some additional information, e.g. the mean BC concentration and the number of fractions of internally/externally mixed particles and coated particles provided by SP2. How was it ensured that the purely externally mixed non-absorbent particles were used in the calculation of the RRI?

Line 216: Can the authors estimate the fraction of the light scattering size distribution that is covered.

Lines 22ff: The uncertainty of the transfer function is covered by the H fitting, since the transfer function is a system function and relatively stable and also covered by a DMA calibration with size standards. How are other influences taken into account, e.g. uncertainties in sheath air flow or CPC counting efficiency?

Line 225: HW not defined

Line 227 to 229: Does this mean that the additional broadening by the H distribution function is 1.073?

Lines 230,231: Can the authors give more details about the uncertainty analysis?

Lines 243, 244: How can it be ensured in a mixed aerosol that BC containing particles are excluded and how big would the error be if small amounts of BC affect the measurement?

---

## Author Comment (AC1) · 22 Apr 2019

Thanks for the reviewer's helpful suggestions! The comments are addressed point-by-point. The response and the revised manuscript are in the supplement.

Please also note the supplement to this comment:
https://www.atmos-meas-tech-discuss.net/amt-2018-399/amt-2018-399-AC1-supplement.zip

---

## Author Comment (AC2) · 22 Apr 2019

Thanks for the reviewer's helpful suggestions! The comments are addressed point-by-point. The revised manuscript and the reply are shown in the supllmentary files.

Please also note the supplement to this comment:
https://www.atmos-meas-tech-discuss.net/amt-2018-399/amt-2018-399-AC2-supplement.zip

---

## Author Response (AR1)

Response to reviewer#1

Thanks for the reviewer's helpful suggestions! The comments are addressed point-by-point and responses are listed below.

**Comments:** In this manuscript, the authors introduced a method to retrieve the real part of refractive index (RRI) of ambient aerosols from the measurements of the scattering intensities of size-selected aerosol particles by the combination of the differential mobility analyzer (DMA) and the single particle soot photometer (SP2). The authors pointed out that retrieval of the size-resolved RRI of ambient aerosols is the innovation of this paper in comparison with the measurements of the total equivalent aerosol RRI or aerosol RRI at a given diameter in previous studies. It is a meaningful topic to measure the real part of the complex refractive index of ambient aerosol particles. However, there are still some important issues should be considered before it is publishable.

**Reply:** We thank the anonymous reviewer's comments and suggestions.

**Comments:** 1. The authors pointed out that "there might be significant variations in the aerosol RRI for aerosols of different diameter because the aerosol RRI is highly related to the aerosol density and chemical components...information of the size-resolved aerosol RRI can help to study the chemical information and the aging process of aerosols among different diameters". However, the results of the size-resolved RRI of the ambient aerosols do not show significant variations among different diameters. The authors should give explanations.

**Reply:** Thanks for the comments. The ratios of aerosol chemical components are different for different diameters, which might lead to significant variations in aerosols RRI for different diameters. The aerosol chemical component, which is not measured in our study, may not vary significantly for different diameters during the test.

The following discussion would demonstrate that the ambient aerosol RRI can vary significantly among different diameter. The aerosol RRI are estimated by using the measured size-resolved main chemical components of the ambient aerosol from Liu et al. (2014) in the North China Plain. The measured data is shown in fig. R1. From fig. R1, the ambient aerosols in the North China Plain are mainly composed of  $NH_4^+$ ,  $NO_3^-$ ,  $Ca^{2+}$ ,  $SO_4^{2-}$ . These chemical components varies among different diameters. The aerosol RRI were estimated using these measured data and the method of Stelson (1990), and shown in fig. R1(b) in dotted black line. Results show that the aerosols RRI for different diameter change significantly between 1.46 and 1.59.

In our study, the DMA-SP2 system measures the aerosol diameter range between 200 and 450 nm. The content, to which the chemical components may change, is not well known due to the lack of size-resolved aerosol chemical information. More measurements were necessary to study the characteristics of the size-resolved RRI. However, this study mainly focus on the method of measuring the size-resolved RRI.

The reviewer gives a new insight into our future work. We also add some revisions accordingly at section 4.1.

**Figure R1.** The average size distributions of the particle chemical composition during the a field campaign (Liu et al., 2014). Panel (a) shows the mass concentration of eight main species and panel (b) shows the relative mass fractions in individual impactor stage. The dotted black line in (b) shows the estimated aerosol RRI.

**Comments:** 2. The size-resolved aerosol RRI is retrieved based on the Mie scattering theory at a given particle diameter. What is the effect of the imaginary part of the complex refractive index on the retrieval?

**Reply:** Thanks for the comments. In our study, the effects of the imaginary part on the retrieval are not considered because we only select the BC free aerosols for this study.

At the same time, we added some discussions in section 4.2.2 on the uncertainties when the aerosols contain a small amount of BC cores that is below the detection threshold of SP2. Monte Carlo simulations were applied to investigate the influence of the BC core on the retrieved ambient aerosol RRI. We found that these particles can lead to less than 0.02 overestimation of the aerosol RRI for most of the conditions.

There are some organic components that may weakly absorb the light intensity. The imaginary part of the refractive index at a given wavelength  $\lambda$  ( $k_{\lambda}$ ) of the BrC can be calculated as:

$$k_{\lambda 1} = k_{\lambda 2} \times (\frac{\lambda 2}{\lambda 1})^w \tag{1},$$

Where w is defined by mass of BC to organic aerosol ratio (R) (Saleh et al., 2015)with:

$$w = \frac{0.21}{R + 0.07} \tag{2}.$$

Based on the work of Saleh et al. (2015), the  $k_{550}$  can be expressed as:

$$k_{550} = 0.016 \times \log_{10}(R) + 0.04 \tag{3}.$$

The values R ranges between 0.09 and 0.35 for different types of aerosols (Saleh et al., 2015). Based on equation (8), (9) and (10), the  $k_{1024}$  ranges between 0.01 and 0.024. The maximum value 0.024 is used for further analysis.

The uncertainties of the retrieved RRI when ignoring the effect of BrC are analyzed. Firstly, The scattering light intensity at a given diameter with a refractive index of 1.46 + 0.024i is calculated using the Mie model. Then the corresponding RRI are retrieved with given diameter and the calculated light intensity. The retrieved aerosol RRI for different aerosol diameter are shown in fig. 7(b). For the light absorbing particles, their scattering light intensity is smaller than that of the pure

scattering particles with the same diameter and RRI. Therefore, the retrieved aerosol RRI is underestimated for most of the conditions. The differences between the given RRI value (1.46) and retrieved RRI value are lower than 0.006 for all of the diameters as shown in fig. 7(b) in the manuscript. The BrC component have little influence on the retrieved aerosol RRI.

The discussions of influence of BrC on the retrieving aerosol RRI are added in section 4.2.3 in the manuscript.

**Comments: 3.** The impact of non-sphericity of ambient aerosols on the light scattering cannot be neglected, especially for dry particles. The authors should also discuss the uncertainties introduced by the sphericity assumption based on the Mie theory.

**Reply:** Thanks for the comments. A lot of closure studies between the measured and calculated aerosol optical properties validate the non-sphericity of the ambient continental (Chen et al., 2014; Ma et al., 2014; Ma et al., 2011; Wex et al., 2002). Based on these studies, it is applicable that these particles are spherical for accumulation mode aerosols.

**Comments:** More specific comments:1. Some details of the method to retrieval real part of the refractive index based on the Mie scattering theory should be added.

**Reply:** Thanks for the comments. We added some descriptions about Mie scattering theory and the method at section 3.1. The method of retrieving the RRI are summarized as follows: (1) measuring the scattering peak height H values at a given diameter; (2) transferring the H into to the light scattering intensity S as denoted in equation 6 in the manuscript by the established relationship from calibration; (3) calculating the refractive index using equation 6 with the given diameter and S.

**Comments:** 2. Lines 57-58: the authors pointed out that "Up to now, there is no information in the literature of the size-resolved ambient aerosol RRI over the

diameter range between 200nm and 500nm...". However, the individual particle analysis combining scanning and transmission electron microscopy (SEM and TEM) have been widely used to derive size-resolved information of the complex refractive index of atmospheric aerosol particles (e.g., in the size range from 100 nm up to 50 µm in diameter) (Ebert et al.,2002, 2004; Kandler et al., 2007).

**Reply:** Thanks for the comments. We revised the manuscript correspondingly. We thank the reviewer for providing us alternative methods to measure the aerosol RRI. The information was summarized and added in the introduction part.

**Comments:** 3. Section 4.1: The field measurements were carried out at the AERONET BEIJING\_PKU station. The results should be compared with the AERONET retrievals considering that the size-resolved RRI of the ambient aerosols doesn't show significant variation among different diameters.

**Reply:** Thanks for the comments. The comparison of the results are shown here but not added in the manuscript. The used RRI from AERONET and our proposed method are these results from 9th to 19th, Match in 2018. Results shown that the RRI measured by the two methods is not correlated with each other. The RRI retrieved from AERONET is the results from column averaged value. At the same time, the aerosol optical properties measured by AERONET are at ambient RH. The RH in the mixed layer increases with height, and reaches larger than 90% frequently in the North China Plain (Kuang et al., 2016; Zhao et al., 2017). When the ambient RH is high, the aerosol takes water and then gets hygroscopic growth. The corresponding aerosol RRI should be lower than that of the dried aerosol particle. The measured RRI in our methods are those of the dried aerosol with RH lower than 40%. Therefore, we don't think it necessary to present the comparison of the RRI measured by our method and the RRI from the AERONET retrievals.

Figure R3. The comparison of the measured RRI by SP2 and AERONET.

**Comments:** 4. Lines 232-233: "For most ambient aerosols, the RRI ranges from 1.4 to 1.5 ...". Some researches have reported the values around 1.53~1.57 for the RRI of most of dry components of atmospheric aerosols, and higher values for RRI of black carbon (BC) component (Xie et al., 2017). The authors should demonstrate their results with other measurements.

**Reply:** Thanks for the comments. Our proposed method focuses on measuring the RRI of the BC free aerosol. Our results show that the ambient BC free aerosol RRI locates around 1.46. More results in another paper in preparation show that the measured RRI can vary a wide range from 1.36 to 1.54. The measured RRI of ambient aerosol is lower than 1.5 because there are many cases that the RRI of organic matter lower than 1.5 (Moise et al., 2015).

At the same time, our method to measure the aerosol RRI is validated by measuring the RRI of ammonium chloride with the RRI of 1.642 as sample aerosol and the corresponding derived size-resolved RRI is  $1.642 \pm 0.02$ .

**Comments:** Typos/Grammar: 1. Line 12 and some other lines in the text: a space should be placed between the number and the unit.

**Reply:** Thanks for the comments. We have changed manuscript correspondingly.

**Comments:**2. Lines 27-29 and some other lines in the text: it is better to use the past tense in review of the literature.

**Reply:** Thanks for the comments. We have changed manuscript correspondingly.

**Comments:** 3. Line 29: please rewrite the sentence "Valenzuela et al. (2018) also reports an uncertainty of 7% with the uncertainties of RRI of 0.1 in RRI." **Reply:** Thanks for the comments. We have changed manuscript correspondingly.

**Comments:**4. Line 58: "the diameter range between 200nm and 500nm where the aerosol scattering coefficients contributes to…". "contributes" should be "contribute". **Reply:** Thanks for the comments. We have changed the paragraph correspondingly.

**Comments:**5. Line 64 and some other lines in the text: "for aerosol of different diameter" should be "for aerosol of different diameters"

**Reply:** Thanks for the comments. We have changed manuscript correspondingly.

**Comments:**6. Line 90: "PNSD" first appears in Section 2.1, but it has not been defined.

**Reply:** Thanks for the comments. We have changed manuscript correspondingly.

**Comments:**7. Lines 99-100 and some other places in the text: the physical quantities "V" and "Zp" should be set in italic in consistent with the equation.

**Reply:** Thanks for the comments. We have changed manuscript correspondingly.

Comments: 8. Line 102: "L" in Eq. (2) has not been defined.Reply: Thanks for the comments. We have changed manuscript correspondingly.

**Comments:**9. Lines 113 and 140: please distinguish the two "C" in Eqs. (5) and (6). **Reply:** Thanks for the comments. We have changed manuscript correspondingly.

**Comments:**10. Lines 150 and 155: "equation (6)" and "equation 6" should be in a uniform format.

**Reply:** Thanks for the comments. We have changed manuscript correspondingly.

**Comments:**11. Line 156: "as that described in section 2.2.1". There is no section 2.2.1 in the manuscript.

**Reply:** Thanks for the comments. We have changed manuscript correspondingly. The true section should be section 2.1.

**Comments:**12. Line 177: "PH0" first appears in Section 3.2, but it has not been defined.

**Reply:** Thanks for the comments. We have changed manuscript correspondingly.

Comments:13. Line 184: "" first appears in Section 3.2, but it has not been defined.Reply: Thanks for the comments. We have changed manuscript correspondingly.

**Comments:**14. Lines 175, 180, 184-185: "fig.2" should be changed into "fig.3". **Reply:** Thanks for the comments. We have changed manuscript correspondingly.

Comments: 15. Line 221: "SP" should be "SP2".

**Reply:** Thanks for the comments. We have changed manuscript correspondingly.

Chen, J., Zhao, C.S., Ma, N., Yan, P. (2014) Aerosol hygroscopicity parameter derived from the light scattering enhancement factor measurements in the North China Plain. Atmos. Chem. Phys. 14, 8105-8118.

Kuang, Y., Zhao, C.S., Tao, J.C., Bian, Y.X., Ma, N. (2016) Impact of aerosol hygroscopic growth on the direct aerosol radiative effect in summer on North China Plain. Atmospheric Environment 147, 224-233.

Liu, H.J., Zhao, C.S., Nekat, B., Ma, N., Wiedensohler, A., van Pinxteren, D., Spindler,

G., Müller, K., Herrmann, H. (2014) Aerosol hygroscopicity derived from size-segregated chemical composition and its parameterization in the North China Plain. Atmospheric Chemistry and Physics 14, 2525-2539.

Ma, N., Birmili, W., Müller, T., Tuch, T., Cheng, Y.F., Xu, W.Y., Zhao, C.S., Wiedensohler, A. (2014) Tropospheric aerosol scattering and absorption over central Europe: a closure study for the dry particle state. Atmospheric Chemistry and Physics 14, 6241-6259.

Ma, N., Zhao, C.S., Nowak, A., Müller, T., Pfeifer, S., Cheng, Y.F., Deng, Z.Z., Liu, P.F., Xu, W.Y., Ran, L., Yan, P., Göbel, T., Hallbauer, E., Mildenberger, K., Henning, S., Yu, J., Chen, L.L., Zhou, X.J., Stratmann, F., Wiedensohler, A. (2011) Aerosol optical properties in the North China Plain during HaChi campaign: an in-situ optical closure study. Atmos. Chem. Phys. 11, 5959-5973.

Moise, T., Flores, J.M., Rudich, Y. (2015) Optical properties of secondary organic aerosols and their changes by chemical processes. Chemical Reviews 115, 4400-4439. Saleh, R., Marks, M., Heo, J., Adams, P.J., Donahue, N.M., Robinson, A.L. (2015) Contribution of brown carbon and lensing to the direct radiative effect of carbonaceous aerosols from biomass and biofuel burning emissions. Journal of Geophysical Research: Atmospheres 120, 10,285-210,296.

Stelson, A.W. (1990) Urban aerosol refractive index prediction by partial molar refraction approach. Environ.sci.technol 24:11, 1676-1679.

Wex, H., Neusüß, C., Wendisch, M., Stratmann, F., Koziar, C., Keil, A., Wiedensohler, A., Ebert, M. (2002) Particle scattering, backscattering, and absorption coefficients: An in situ closure and sensitivity study. Journal of Geophysical Research: Atmospheres 107, LAC 4-1-LAC 4-18.

Zhao, G., Zhao, C., Kuang, Y., Tao, J., Tan, W., Bian, Y., Li, J., Li, C. (2017) Impact of aerosol hygroscopic growth on retrieving aerosol extinction coefficient profiles from elastic-backscatter lidar signals. Atmospheric Chemistry and Physics 17, 12133-12143.

**Response to reviewer#2**

Thanks for the reviewer's helpful suggestions! The comments are addressed point-by-point and responses are listed below.

**Comment 1:** General comments: The present manuscript describes a method for deriving the real part of the refractive index by means of a differential mobility selector (DMA) and scattering intensities measured with a SP2. The derivation of the real part of the refractive index of a quasi-mono disperse aerosol is not completely new. What is new, however, is the application with the use of the SP2, which in a unique way can also determine the mixing state of the aerosol within certain limits. This ensures that errors caused by unknown imaginary parts of the refractive index are minimized. The method shown is limited to non-absorbent particles.

**Reply 1:** We agree with the anonymous reviewer's comments.

**Comment 2:** The reviewer thinks that the current limitations and consequences have not been adequately presented. In particular, a consideration of the uncertainties in violation of the restrictions (weakly absorbing particles) is lacking.

**Reply 2:** Thanks for the comments. The reviewer provides a good view in uncertainties analysis of our proposed method.

There are some brown carbon (BrC) that absorb the light intensity in the near infrared range. The imaginary part of the refractive index at a given wavelength  $\lambda$  ( $k_{\lambda}$ ) of the BrC can be calculated as:

$$k_{\lambda 1} = k_{\lambda 2} \times \left(\frac{\lambda 2}{\lambda 1}\right)^w \tag{1},$$

Where w is defined by mass of BC to organic aerosol ratio (R) (Saleh et al., 2015)with:

$$w = \frac{0.21}{R + 0.07} \tag{2}.$$

Based on the work of Saleh et al. (2015), the  $k_{550}$  can be expressed as:

$$k_{550} = 0.016 \times \log_{10}(R) + 0.04 \tag{3}$$

The values R ranges between 0.09 and 0.35 for different types of aerosols (Saleh

et al., 2015). Based on equation (8), (9) and (10), the  $k_{1024}$  ranges between 0.01 and 0.024. The maximum value 0.024 is used for further analysis.

The uncertainties of the retrieved RRI when ignoring the effect of BrC are analyzed. Firstly, The scattering light intensity at a given diameter with a refractive index of 1.46 + 0.024*i* is calculated using the Mie model. Then the corresponding RRI are retrieved with given diameter and the calculated light intensity. The retrieved aerosol RRI for different aerosol diameter are shown in fig. 7(b). For the light absorbing particles, their scattering light intensity is smaller than that of the pure scattering particles with the same diameter and RRI. Therefore, the retrieved aerosol RRI is underestimated for most of the conditions. The differences between the given RRI value (1.46) and retrieved RRI value are lower than 0.006 for all of the diameters as shown in fig. 7(b) in the manuscript. The BrC component have little influence on the retrieved aerosol RRI.

We added some discussions in section 4.2 on the uncertainties when the aerosols contain a small amount of BC cores that are below the detection threshold of SP2. Monte Carlo simulations were applied to investigate the influence of the BC core on the retrieved ambient aerosol RRI. These particles can lead to less than 0.02 overestimation of the aerosol RRI for most of the conditions. More details are shown in **Reply 4**.

Some corresponding discussions were added in section 4.2.3 and 4.2.4 in the text.

**Comment 3:** In laboratory experiments, as shown with ammonium sulfate and ammonium chloride, this is easily possible. The application to a complex ambient aerosol, on the other hand, was not treated sufficiently. The example shown in chapter 4.1 shows results of measurements in Beijing, where a complex mixed aerosol is present (that measurement place was characterized as urban roadside; line 96).

**Reply 3:** We thank the anonymous reviewer's comments and suggestions. In this work, we mainly focus on the method of measuring the aerosol size-resolved real part of the refractive index (RRI). The ammonium sulfate is used for calibration, and the ammonium chloride is used for validation. These studies are easy but power

demonstrations that our proposed is applicable of measuring the BC-free aerosol. More results about field measurements using this method can refers to another work at https://www.atmos-chem-phys-discuss.net/acp-2019-250/.

**Comment 4:** An error estimation is missing: a) what happens with weakly absorbing organic droplets, b) what happens with internally mixing particles with a small soot core when the incandescence signal is below the detection threshold of SP2. How large are the expected errors in the real part of the refractive index?

**Reply 4:** We do appreciate the comments. The reviewer provides a good view in uncertainties analysis of our proposed method. We added discussions on this point in section 4.2.

For the organic droplets, the light absorption of these components is ignorable as detailed in reply 2.

There are some particles with a small soot core and the incandescence signals are below the detection threshold of SP2. The derived aerosol RRI should be influenced by small soot core. Uncertainties might be resulted when deriving the RRI for these BC-contained aerosols.

For the BC-contained aerosol, Aquadag soot particles with effective density of 1.8 g/cm3 is used to determine the lower limit of the BC particle diameter when the incandescence signals can be detected by SP2. The calibration procedure is conducted the same as that of the ammonia sulfate in the manuscript. The diameters (Dp) of the aerosol passing through the DMA are manually changed from 60 to 400 nm with a step of 20 nm. The relationships between the measured incandescence signal height and the Dp are shown in fig. R2. From fig. R2, we conclude that our SP2 is not capable of measuring the Aquadag soot particles lower than 80 nm.

**Figure R2.** The calibrated relationship between the incandescence peak height and the BC diameter for both the incandescence high gain channel and the incandescence low gain channel.

We derived the aerosol equivalent refractive index when the aerosol have BC cores lower than 80 nm with two steps. 1, the scattering strength of the BC-containing aerosols are calculated based on Mie scattering theory with a given core and total diameter. 2, the scattering strength are used to deriving the equivalent refractive index with assuming that the BC-containing aerosols are pure scattering aerosols.

Monte Carlo simulations were applied to investigate the influence of the BC core on the retrieved ambient aerosol RRI. Firstly, the aerosol diameter are first chosen between 200 nm and 500 nm. Then the core diameter are random determined lower than 80 nm. The core diameters flow the log-normal distribution with the mean core diameter of 120 nm (Raatikainen et al., 2017). When calculating the scattering strength, the complex refractive index of the core 1.8+0.54i (Zhao et al., 2018) is used. The complex refractive of the shell adopts the measured mean values (1.46+0i) during the field measurements. The scattering strength can be calculated with the above information and the Mie scattering Model. Then with the calculated scattering strength, the equivalent real part of the refractive index (RRI) can be derived with assuming that the aerosols are pure scattering aerosols. If the core diameter is zero, the derived aerosol equivalent aerosol RRI should be 1.46.

For each aerosol diameter, the Monte Carlo simulations were conducted for

10000 times. Figure. R3 gives the retrieved aerosol equivalent RRI at different diameters. Results show that the retrieved aerosol equivalent RRI are larger than 1.46 for all of the given aerosol diameters. When the aerosols have BC core, the scattering strength are larger than that of pure scattering aerosols with the same aerosol diameter. The derived mean equivalent RRI tend to be closer to 1.46 when the aerosol diameter is larger, where the BC core contributes less and the influence of the BC core are smaller. The derived mean aerosol equivalent RRI is 1.47 and 1.462 at 200 nm and 500 nm respectively. At the same time, the uncertainties associated with the equivalent RRI are larger when the aerosol diameter is smaller. We conclude that the uncertainties associated with BC core are smaller than 0.01 when the aerosol diameter are larger than 250 nm. The maximum of the difference of the derived RRI is 0.02.

The above corresponding discussions were added in the uncertainties analysis part in section 4.2.2 of the manuscript.

---

## Author Response (AR2)

**Comment:** The components of ambient aerosols are complex. The authors have pointed out that there might be significant variations in the RRI for aerosols of different diameters because the aerosol RRI is highly related to the aerosol chemical component. In this manuscript, the results of the derived RRI of the ambient aerosols in the range from 200 nm to 450 nm do not show obvious variations among different diameters. I suggest the authors add some cases to show variations in the size-resolved RRI which will be more convincing.

**Reply:** Thanks for the comment. Fig. R1 shows the measured RRI values at different diameters during the field measurement on 23 June, 2018, at a suburban site Taizhou (119º57'E, 32º35'N), which lies in the south end of the Jianghuai Plain in the central Eastern China. From fig. R1, the measured mean RRI varies significantly among different diameters from 1.47 at 198 nm to 1.54 at 450 nm.

[Figure]

**Figure R1.** Measured RRI among different diameters.